# Coupled Electrostatic and Hydrophobic Destabilisation of the Gelsolin-Actin Complex Enables Facile Detection of Ovarian Cancer Biomarker Lysophosphatidic Acid

**DOI:** 10.3390/biom13091426

**Published:** 2023-09-21

**Authors:** Katharina Davoudian, Shayon Bhattacharya, Damien Thompson, Michael Thompson

**Affiliations:** 1Department of Chemistry, University of Toronto, 80 St. George Street, Toronto, ON M5S 3H6, Canada; k.davoudian@mail.utoronto.ca; 2SSPC—The Science Foundation Ireland Research Centre for Pharmaceuticals, V94 T9PX Limerick, Ireland; shayon.bhattacharya@ul.ie; 3Department of Physics, Bernal Institute, University of Limerick, V94 T9PX Limerick, Ireland

**Keywords:** ovarian cancer, lysophosphatidic acid, gelsolin, actin, lipid-protein interaction, predictive molecular modelling

## Abstract

Lysophosphatidic acid (LPA) is a promising biomarker candidate to screen for ovarian cancer (OC) and potentially stratify and treat patients according to disease stage. LPA is known to target the actin-binding protein gelsolin which is a key regulator of actin filament assembly. Previous studies have shown that the phosphate headgroup of LPA alone is inadequate to bind to the short chain of amino acids in gelsolin known as the PIP_2_-binding domain. Thus, the molecular-level detail of the mechanism of LPA binding is poorly understood. Here, we model LPA binding to the PIP_2_-binding domain of gelsolin in the gelsolin-actin complex through extensive ten-microsecond atomistic molecular dynamics (MD) simulations. We predict that LPA binding causes a local conformational rearrangement due to LPA interactions with both gelsolin and actin residues. These conformational changes are a result of the amphipathic nature of LPA, where the anionic phosphate, polar glycerol and ester groups, and lipophilic aliphatic tail mediate LPA binding via charged electrostatic, hydrogen bonding, and van der Waals interactions. The negatively-charged LPA headgroup binds to the PIP_2_-binding domain of gelsolin-actin while its hydrophobic tail is inserted into actin, creating a strong LPA-insertion pocket that weakens the gelsolin–actin interface. The computed structure, dynamics, and energetics of the ternary gelsolin–LPA–actin complex confirms that a quantitative OC assay is possible based on LPA-triggered actin release from the gelsolin-actin complex.

## 1. Introduction

Ovarian cancer (OC) is the most fatal and fifth most diagnosed gynaecological cancer, with a five-year survival rate of less than 30%, making it the eighth leading cause of all deaths due to cancer [1,2]. Despite the ongoing development of new therapeutic options, the treatment of OC is hampered by tumour recurrence and severe drug resistance [1]. The disease is often termed a “silent killer” because the early stages of the cancer are considered to be asymptomatic, making it extremely challenging for diagnosis and prognosis. However, studies show that most early stage OC patients have in hindsight experienced at least one symptom. These symptoms are nonspecific and can be characteristic of other conditions. They are typically abdominal, gastrointestinal, and/or urinary related, such as pelvic pain, bloating, or trouble eating [3,4,5,6,7]. The consensus that early OC is silent, coupled with a lack of screening options and early stage symptom awareness, contributes to the common late diagnosis of OC.

About 75% of OC patients are diagnosed very late, in stages III or IV [8], which significantly reduces the five-year survival rate, from above 90% in stage I to around 20–30% in stage IV. Epithelial OC, the most common OC subgroup, has the lowest survival rate [8,9,10]. Current OC diagnostic tests are limited by a substantial risk of false negatives, as well as false positives [11]. Diagnosis for OC includes a physical exam (which can be challenging due to tumour depth [12]), analysis of tissue biopsies, an unreliable blood test (detailed below), and costly imaging techniques such as transvaginal ultrasound (TUS), computerised tomography (CT) scans, and magnetic resonance imaging (MRI) [7,13]. Typically, OC is diagnosed by histological analysis of tissue biopsies, which is a reliable yet invasive method [7,14]. To improve prognosis, clinical screening and symptom awareness of early stage OC is critical.

Cancer antigen 125 (CA-125) is the only OC blood serum biomarker that can be clinically tested, with a sensitivity of 75–82% and specificity of 67–76% [15,16]. As CA-125 can also be present in benign conditions and other cancers and is only increased in about 50% of early stage OC patients, this inconclusive blood test for OC must be combined with the expensive and time-consuming imaging techniques [7,10,15]. Blood levels of CA-125 increase as OC advances and it is present in about 92% of advanced-stage patients, making it more useful for monitoring disease progression during treatment [10,15].

It is clear, therefore, that detecting OC requires a reliable and widely accessible blood test that is inexpensive and rapid with high sensitivity and specificity. The diagnostic test for OC must detect an early stage biomarker (or multiple biomarkers) to enable early treatment when the five-year survival rate is high.

We have developed an experimental fluorescence microscopy assay for sensing the OC biomarker lysophosphatidic acid (LPA) in serum [17]. LPA dissociates the gelsolin-actin complex, which is immobilised on silica gel via a histidine tag. Improved sensor surface packing and sensitivity were achieved with gelsolin(1–3) instead of the full gelsolin protein, which is a homologous dimer containing the identical halves of gelsolin(1–3) and gelsolin(4–6). The sensor measures levels of labelled actin in solution which in turn indicates LPA concentration in blood [17]. The technique is reliable and accessible, and its further development and adaption for sensing and diagnostics applications will be accelerated by understanding the underlying molecular mechanism by which LPA dissociates the gelsolin-actin complex.

LPA is a bioactive lysophospholipid consisting of a glycerol residue with one free hydroxyl group, one hydroxyl esterified with a phosphate group, and the third esterified with an unsaturated fatty acid (see Figure 1). LPA has varying biological functions depending on its structure. It can induce intracellular signalling that causes cell migration, proliferation, and cytoskeletal changes by interacting extracellularly with its six G protein-coupled receptors [18,19,20]. LPA is synthesised both intra and extracellularly, but its intracellular synthesis is believed to be for phospholipid production [19,20]. Outside the cell, LPA can be derived from membrane phospholipids or lysophosphatidylcholine [19,20,21].

While LPA is associated with diverse biological functions, it also plays a crucial role in many diseases, including cancers and Alzheimer’s and Parkinson’s neurodegeneration [20]. In OC, previous studies have reported that abnormal LPA expression can induce irregular signalling and expression of its receptors, initiating disease onset and progression [15,22]. Xu et al., stated that LPA is overexpressed in 90% of stage I OC [23], indicating that LPA levels can potentially distinguish between healthy individuals and those with cancerous or benign ovarian tumours. However, some studies disagree with this finding [24,25] and further research is required to confirm the specificity of LPA for early stage OC. Nevertheless, an LPA assay can be useful for monitoring OC as other works reported that LPA levels increase with the advancement of OC. This suggests that LPA detection in plasma or serum can identify the stage of the disease [10,15,17], which is crucial for stratifying patients and developing personalised treatments appropriate to the disease state. While the average healthy LPA concentration is 1.2 µM, levels can surge from 1.3 to 50 µM depending on the OC stage [17,23,26].

LPA is an intracellular modulator of gelsolin-actin binding [27]. Gelsolin is a large multidomain protein of ~85 kDa [28]. Its domains 1–3 (known as the N-terminal half) and 4–6 (the C-terminal half) are homologous [29,30]. To match the experimental sensor [17] and for computational expediency, we use gelsolin(1–3) as our model gelsolin protein. Gelsolin is a cytoskeleton regulator, modulating actin severance and polymerisation. Gelsolin binds with actin quickly, but separation occurs slowly [30], suggesting a structural reorganisation of the complex prior to gelsolin decoupling from the actin filament. Following actin severing, gelsolin may adhere to actin as a capping protein, preventing the elongation of actin filaments. When gelsolin uncaps, the non-barbed end of actin is exposed, allowing actin to polymerise [30].

A study by Goetzl et al., showed that PIP_2_ is a competitive inhibitor of LPA-gelsolin binding [31]. The region where PIP_2_ binds, residues 135–169 in gelsolin(1–3), is called the PIP_2_-binding domain [28]. One of the functions of PIP_2_ is to inhibit gelsolin severing of actin [32]. Like PIP_2_, LPA binds to the PIP_2_-binding domain of gelsolin and is also efficient in separating gelsolin-actin complexes [28]. The similar binding mechanism of PIP_2_ and LPA with gelsolin [31] is likely due to their similar structures, as both molecules are amphipathic (see Figure 1).

The PIP_2_-binding domain (^135^KSGLKYKKGGVASGFKHVVPNEVVVQRLFQVKGRR^169^) [28] of gelsolin includes basic residues lysine and arginine that are protonated at physiological pH. These positively charged residues may be key for interacting with the negative phosphate headgroup of PIP_2_. However, a study by Feng et al., showed that the polar headgroup of PIP_2_ alone is inadequate to bind gelsolin and compete with competitive inhibitors of gelsolin; the study revealed that both its polar headgroup and nonpolar aliphatic chains may be essential for binding [32]. As LPA is structurally similar to PIP_2_, this suggests that LPA binding to gelsolin may involve a combination of different interactions (electrostatic salt bridges, hydrogen bonds, and hydrophobic van der Waals contacts). The amphipathic nature of LPA was used by Chen et al., to connect quantum dots (QDs) with graphene nanosheets (GNs), with the polar headgroup of LPA interacting with positively charged guanidine groups on the QDs and the GNs binding the nonpolar tail of LPA [22].

Experimentally, the overall binding affinity between gelsolin and LPA is reported to be high (Kd = 6 nM) [31], while a portion of the PIP_2_-binding domain (residues 150–169, called P2) has a variable affinity to LPA depending on salt concentration (0.92–24 μM) and temperature [28]. This suggests that other structural domains of gelsolin may further contribute to LPA-gelsolin binding [17]. Compared to whole gelsolin, gelsolin(1–3) may have a different binding mechanism to LPA. This is because, in addition to the PIP_2_-binding domain, PIP_2_ also binds with residues 621–634 (called P3) in gelsolin(4–6) [32]. To the best of our knowledge, binding affinity between LPA and gelsolin(1–3) has not been reported to date, beyond the inference of binding from the experimental sensing measurements [17].

In addition to PIP_2_, gelsolin activity is also regulated by binding to eight calcium ions (Ca^2+^) [29,30] that are grouped into two types: two Type 1 calcium ions that bind both gelsolin and actin and six Type 2 calcium ions that only coordinate to gelsolin [33]. Type 2 Ca^2+^ enables activation and binding to actin through structural rearrangements in gelsolin [33]. Micromolar concentrations of calcium ions and slight acidity activate gelsolin [32] through conformational changes that expose the actin-binding sites on gelsolin that sever and cap actin [30,34]. While whole gelsolin is Ca^2+^-dependent, calcium ions influence the gelsolin component halves differently; gelsolin(4–6) is Ca^2+^-dependent but gelsolin(1–3) is not [30,35,36]. Gelsolin(1–3) can sever and cap actin without the need for calcium ions, indicating that calcium is not required for gelsolin(1–3) to bind with actin [30,36].

In this work, we model the molecular mechanism by which LPA acts as an early stage OC biomarker. We study its binding efficiency to the PIP_2_-binding domain of gelsolin(1–3) and reveal the structural rearrangements, interactions, and thermodynamic stabilities of the LPA-bound gelsolin(1–3)-actin complex, through extensive molecular dynamics (MD) simulations (Figure 2). As gelsolin has two homologous regions, the N-terminal half (domains 1–3) was used in our previous work [17] and therefore chosen for modelling here. Our models predict that LPA interacts with actin and the PIP_2_-binding domain of gelsolin(1–3) simultaneously through combined electrostatic and steric interactions that weaken gelsolin(1–3) binding to actin, showing propensity of the two proteins to dissociate from each other in the presence of LPA.

## 2. Materials and Methods

### 2.1. Source of Protein and LPA Structures

An X-ray crystal structure (resolution 3Å) of the human gelsolin(1–3)-actin complex with ATP and five calcium ions was obtained from the Protein Data Bank (PDB code 3FFK) [29]. Lysophosphatidic acid (LPA) was acquired from an X-ray crystal complex structure of autotaxin, tauroursodeoxycholic acid, and LPA (PDB code 5DLW) [37]. To examine the effect of LPA on the gelsolin(1–3)-actin complex under physiological conditions, the ionisation of LPA was determined using the Schrödinger Maestro package [38] (see Appendix A). LPA was predicted to have a −2e negative charge at pH 7.4, as shown in Figure 1). Oleoyl-L-α-lysophosphatidic acid is modelled here as our previous experimental measurements [17] used the 18:1 LPA isomer purchased from Sigma Aldrich (Oakville, ON, Canada).

### 2.2. Macromolecular Docking

Blind unbiased macromolecular docking of LPA to the gelsolin(1–3)-actin complex was performed to determine if LPA would dock to the PIP_2_-binding domain of gelsolin(1–3). As the unbiased docking did not produce energetically favourable models of LPA bound to the PIP_2_-binding domain, biased docking was performed, restricting LPA to bind to the PIP_2_-binding domain. For this work, the longest reported PIP_2_-binding domain was chosen, covering gelsolin residues 135–169 [28]. Four codes were used to perform the docking calculations: AutoDock 4 [39], AutoDock Vina [40] via UCSF Chimera [41], and HDOCK [42]. MD simulations were performed with the most energetically favourable LPA-bound models calculated from the three docking methods.

### 2.3. Molecular Dynamics Simulations

The gelsolin(1–3)-actin protein complex with ATP and calcium ions was represented by CHARMM36m forcefield parameters [43] while the CHARMM General Force Field (CGenFF) [44] was used to model the topology and parameters of LPA. Each complex was solvated in a large cubic water box with water molecules represented by the CHARMM-modified TIP3P [45] water model. The minimum distance between the complex and the water box edge was kept at least 20 Å. Background Na^+^ or Cl^–^ counterions were added to neutralise any formal charge in the complex and NaCl salt was added at a physiological concentration of 0.15 M.

All MD simulations were carried out using the Gromacs 2018 package [46] with an integration time step of 2 fs implemented in the leapfrog integrator [47] with bond lengths to hydrogen constrained using the LINCS [48] (protein and ATP) and the SETTLE [49] (water) algorithms. Snapshots were saved every 2 ps. Long-range electrostatics were treated by the Particle Mesh Ewald (PME) method [50]. Protein and non-protein molecules (water and ions) were coupled separately to an external heat bath (310 K) with a coupling time constant of 1 ps using the velocity rescaling method [51]. All systems were energy minimised then thermalized over 100 ps and equilibrated for 1 ns in constant volume NVT ensemble followed by another 1 ns of NPT equilibration with the reference pressure at 1 bar and a time constant of 4 ps using the Berendsen barostat [52]. The long production runs of equilibrated molecular dynamics were carried out in the constant pressure NPT ensemble using the Parrinello–Rahman barostat [53].

MD simulations with the docked model of AutoDock Vina showed that LPA left the gelsolin(1–3)-actin-binding pocket after just 6 ns of dynamics. The other two docked models, from HDOCK and AutoDock 4, were simulated for 2 microseconds each. As the complex modelled with AutoDock 4 showed superior binding affinity (discussed later) of LPA to gelsolin(1–3)-actin compared to the other model, this complex was also simulated in the absence of calcium ions to study the effect of calcium ions on the molecular recognition. Additional control simulations were performed without bound LPA, both with and without the bound calcium ions. Henceforth, we use the following naming convention for all model MD simulations, unless otherwise specified: AutoDock Vina model = Dock 1, HDOCK model = Dock 2, AutoDock 4 model = Dock 3, model without LPA = LPA-free, model without LPA and calcium ions = LPA-CAL-free, and model without calcium ions = CAL-free.

### 2.4. Model Analysis

All trajectories were analysed with Visual Molecular Dynamics (VMD) package [54]. The number of hydrogen bonds and the hydrogen-bond occupancy were calculated using a cut-off distance of 3.5 Å and cut-off angle of 30°. Gromacs tools were used to calculate root mean square deviation (RMSD), root mean square fluctuation (RMSF), and secondary structure features of the gelsolin(1–3)-actin complex, and to compute interaction energies between LPA and the complex. The fraction of native contacts (*Q*) was calculated using the MDtraj python library. *Q* was based on the definition by Best, Hummer, and Eaton [55] given by Equation (1):(1)$$Q\left(x\right)=\frac{1}{N}\sum_{\left(i,j\right)}\frac{1}{1+exp\left[\beta (r_{ij}(x{)}^{0}-{\lambda r}_{ij}^{0}\right]},$$
where *N* represents the set of all pairs of heavy atoms (*i*, *j*) that are in contact if their distance is less than 5 Å and they are separated by at least 3 residues. The distance between heavy atoms *i* and *j* in the conformation (*x*) sampled at time t is represented by r*_ij_*(*x*), while the distance in the starting structure at time 0 is r*_ij_*^0^. The smoothing parameter, β, is taken as standard to be 5 Å^−^^1^, and λ is a factor that describes fluctuations when the contact is formed, with a standard value of 1.8 for the all-atom model. The LPA binding energies were calculated using Molecular Mechanics Poisson–Boltzmann surface area (MM/PBSA) method [56] implemented in the g_mmpbsa package [57]. Protein tertiary contact analyses were performed using the CONAN [58] mapping tool. Pairs of residues are in contact if the minimum average distance between the heavy atoms of the two residues is within 5 Å. The intra- and inter-molecular interaction maps (classified as hydrophobic, hydrogen bond, and salt bridge) were generated using a truncation lifetime of 0.5, which counts only contacts with probabilities greater than 50%.

## 3. Results and Discussion

### 3.1. LPA Binding Is Not Restricted to the PIP_2_-Binding Domain of Gelsolin(1–3)

The starting structures show LPA docked in three different binding modes on the gelsolin(1–3)–actin protein complex, named the Dock 1–3 models (Figure 3). In Dock 2, the structure of LPA is more compact and makes sub-3 Å contacts with residues Gln95, Phe149 of the gelsolin(1–3) protein, and with Gly23, Asp24, Asp25, Ala26, Glu334, Tyr337, Trp340, and Ile341 of the actin protein. In Dock 1, LPA is similarly compact and its phosphate headgroup binds to Ca^2+^ #3. LPA sits within 3 Å of residues Glu92, Ser94, Gln95, Ser147, Phe149, His151 of gelsolin(1–3), and Asp25 of actin. The Dock 3 starting structure shows LPA bound in an elongated state and binds with sub-3 Å contacts to residues Glu92, Ser94, Val145, Ala146, Ser147, Phe149, Lys150, His151 of gelsolin(1–3), and Val9, Asp11, Lys18, Asp24, Asp25, Ala26, Pro27, Trp340, and Ile341 of actin.

The MD simulations reveal that LPA quickly leaves the complex in the Dock 1 model, unbinding after ~6 ns. When LPA loses its hydrogen bond contacts with gelsolin(1–3)-actin at ~6 ns, its electrostatic interactions with the protein become repulsive (Appendix A). Although the negatively charged phosphate headgroup of LPA makes favourable electrostatic interactions with the positively charged Ca^2+^ #3 and Lys142 of gelsolin, LPA is also near the acidic residue Glu92. This electrostatic repulsion, coupled with poor contacts between the nonpolar aliphatic tail of LPA and the polar (Ser94, Gly95, Ser147), basic (His151), and acidic (Asp25) residues, makes LPA binding with the gelsolin(1–3)-actin complex unfavourable in the Dock 1 model. LPA ultimately diffuses into bulk water, indicating that the favourable electrostatic interaction via its headgroup and Lys142 is not sufficient for maintaining LPA-binding as previously reported [32], where the structurally similar PIP_2_ molecule could not bind to gelsolin without its nonpolar tail. LPA binding could therefore be mediated by both van der Waals and electrostatic salt bridge and hydrogen bonding interactions.

During the first nanosecond of molecular dynamics starting from Dock 2, LPA quickly unfolds from its initial compact bound state to attain a more energetically favourable elongated conformation; a similar unfolding occurs for Dock 3. The amphipathic nature of LPA allows its hydrophilic headgroup and hydrophobic tail to participate in distinct non-bonded interactions with complementary sites on the gelsolin(1–3)-actin complex. The LPA binding poses are maintained throughout the two-microsecond dynamics, as evidenced by, for example, the root mean square deviation (RMSD) timelines of LPA (Appendix A).

Both Dock 2 and Dock 3 simulations show the hydrophobic tail of LPA embedding within a tunnel cavity of actin, which we call here the LPA-insertion pocket (see Figure 4). LPA remains bound in the pocket, with the computed structures predicting that LPA binding is not restricted to the PIP_2_-binding domain of gelsolin(1–3). The PIP_2_-binding domain of gelsolin(1–3) may facilitate LPA to bind to the protein, creating an electrostatic encounter complex that places the nonpolar tail in the vicinity of the pocket, leading to LPA interacting with both gelsolin(1–3) and actin, creating conformational rearrangements in both proteins that tighten LPA binding at the expense of weakened gelsolin(1–3)–actin contacts that may trigger actin release measured at experimental timescales [17].

Although LPA is initially docked to slightly different regions in the Dock 2 and 3 starting structures, both MD trajectories show increasing LPA–actin contacts during the two-microsecond runs (see Table 1). For model Dock 2, 90% of all LPA interactions are finally with actin; for model Dock 3 it is 58%. In both models, these actin residues make up the amphipathic LPA-insertion pocket as shown in Figure 4, involving nonpolar residues (Val9, Ala19, Ala22, Gly20, Phe21, Gly23, Pro27, Ile330, Pro332, Tyr337, Trp340, and Ile341) and charged or polar residues (Asp11, Lys18, Asp24, Asp25, Ser145, Arg147, Glu334, Arg335, Ser338, and Ser344). While the actin residues interacting with LPA show more variability during dynamics, the set of amino acid residues in gelsolin(1–3) that interact with LPA remains relatively constant throughout the course of the two-microsecond dynamics. Compared to Dock 2, LPA in Dock 3 interacts more with gelsolin(1–3) residues, mostly belonging to the PIP_2_-binding domain of gelsolin(1–3).

We note that the binding of LPA to the gelsolin(1–3)-actin complex during dynamics of Dock 2 and 3 is similar to the binding of LPA to the enzyme autotaxin (ATX), which produces LPA by converting lysophosphatidylcholine (LPC) to LPA [59]. Salgado-Polo et al., demonstrated through MD simulations that the hydrophobic tail of LPA enters a hydrophobic tunnel in ATX similar to the tunnel cavity in actin predicted in the present work, while its phosphate headgroup undergoes electrostatic interactions with positively charged residues [60]. Our data shows that while LPA in the Dock 2 model is not close to Ca^2+^ #3, the polar headgroup of LPA interacts with two positively charged arginine residues, namely, Arg120 in gelsolin(1–3) and Arg147 in actin (Appendix A). Arginine contains a guanidine group, which is protonated at physiological pH and can participate in electrostatic salt bridges and bidentate hydrogen bonding [61]. Guanidine groups were demonstrated to bind with LPA through electrostatic interactions and hydrogen bonds in a developed LPA assay [22]. In contrast, the headgroup of LPA in Dock 3 is close to Ca^2+^ #3 and makes stabilising interactions with Glu92 as well as polar serines 94 and 147.

### 3.2. LPA-Binding Weakens the Gelsolin(1–3)-Actin Complex

We note that the computed interaction energies have a stronger contribution from Coulomb’s electrostatics compared to van der Waals forces between gelsolin(1–3) and actin for all models except with starting structure Dock 3 (Figure 5A,B). Although all systems have similar contributions from van der Waals interactions between gelsolin(1–3) and actin (approximately −750 to −500 kJ/mol), the electrostatic interactions in Dock 3 weaken over time and are approximately −500 kJ/mol, similar in magnitude to van der Waals, after two microseconds. Interestingly, the Dock 2 and LPA-CAL-free systems have equally strong electrostatic interactions between gelsolin(1–3) and actin (approximately −1500 kJ/mol) during the last 200 ns dynamics, suggesting that the calcium ions do not significantly stabilise the gelsolin(1–3)-actin interactions. The calcium-independent functioning of gelsolin(1–3)-actin was reported in experiments by Nag et al., where the N-terminal half (domains G1–G3 used in this study) of whole gelsolin continues to function regardless of the presence or absence of bound calcium ions [36]. The effect of LPA binding on the gelsolin(1–3)-actin interface is evident from Figure 5B with significantly better gelsolin(1–3)-actin electrostatics in Dock 3 compared with its LPA-free counterpart.

Interaction energies between gelsolin(1–3)-actin and LPA have a similar contribution from van der Waals and electrostatics in Dock 3, while electrostatic interactions are more favourable in Dock 2 (Figure 5C,D). However, the more detailed LPA-protein binding free energy (including solvation effects, see Methods) is more favourable in Dock 3 (Figure 6B). Our models show that, regardless of subtle differences in the initial binding pose, and in the interaction/binding energy calculation method, efficient LPA binding requires contributions from both van der Waals tail-insertion and electrostatic binding energies.

Figure 6 shows the binding free energy predicted from MD simulations of the best-docked poses (Dock 2 and Dock 3). The time-averaged superior binding of LPA to the gelsolin(1–3)-actin complex in Dock 3 (Figure 6B) may originate from two reasons. Inspecting Table 1 we see that, (1) LPA in this model interacts more consistently with the PIP_2_-binding domain of gelsolin(1–3), with (2) a relatively equal distribution of gelsolin(1–3) and actin residues binding LPA. Both the PIP_2_-binding domain of gelsolin(1–3) and the LPA-insertion pocket of actin may synergistically facilitate LPA binding, leading to gradual destabilisation of gelsolin(1–3) binding to actin and release of actin at experimental timescales [17].

### 3.3. Local Conformational Rearrangements in Gelsolin(1–3)-Actin due to LPA Binding Predicts the Efficiency of LPA as an OC Biomarker

Previous reports have suggested that strong polar interactions between regions Lys150–Asn155 of gelsolin(1–3) and Gly23–Val30 of actin are important for gelsolin(1–3)-actin complexation [62,63]. The starting gelsolin(1–3)-actin crystal structure shows that the residue Lys150 in gelsolin(1–3) initially has hydrogen bonds with residues Gly23 and Asp24 in actin. However, in the LPA-bound tri-complex, a local conformational change of the PIP_2_-binding domain of gelsolin(1–3) shifts the positively charged Lys150 residue in gelsolin(1–3) to form strong hydrogen bonds with the negatively charged phosphate headgroup of LPA (Figure 7 and Appendix A). As a result, both the interaction and binding energies between LPA and gelsolin(1–3)-actin improve by approximately 200 kJ/mol (Figure 5 and Figure 6) during the first 200 ns of dynamics (Appendix A). This is also evident from the contribution of Lys150 towards the overall binding energy which is more favourable by around 29.5 kJ/mol in the last 200 ns dynamics (Figure 7). However, there is negligible change in the contribution energy from Ca^2+^ #3, suggesting that the electrostatic interactions of LPA with the protein complex may not be mediated by the calcium ion. Rather, we predict that LPA binding to the gelsolin(1–3)-actin complex is associated majorly with local conformational changes in the gelsolin(1–3)-actin complex. These findings with Dock 3 are in contrast to Dock 2, where Lys150 and LPA do not interact, and LPA does not create conformational rearrangements in the PIP_2_-binding domain.

We further note that the binding pose of LPA in Dock 3 may slightly weaken the secondary structure of the protein complex (Appendix A). The timelines of secondary structures predict more variability in Dock 3 relative to Dock 2 (Appendix A), while most of the structural fluctuations occur in the LPA-CAL-free system (Appendix A). As the number of residues in α-helices and β-sheets decreases, the number of residues in coils increases in Dock 3, indicating that LPA induces conformational changes in the protein. The LPA-free model shows the most stability in its secondary structures, suggesting that the presence of calcium ions and the absence of LPA help maintain the gelsolin(1–3)-actin complex and provide structural stability. Although gelsolin(1–3) maintains its function in the absence of calcium ions and continues to bind with actin [36], the lack of calcium makes the gelsolin(1–3)-actin complex less stable, as the LPA-CAL-free model shows the most decrease in secondary structures.

## 4. Conclusions

Using extensive, long, atomically detailed scale molecular dynamics, we show that the binding of LPA to the gelsolin(1–3)-actin complex is mediated by distinct electrostatic and van der Waals interactions. The strong, large-area interface is created by the charged, polar, and nonpolar residues in the gelsolin(1–3)-actin complex binding to the anionic, polar headgroup, and nonpolar aliphatic tail of LPA, respectively. Our models show that LPA-induced local conformational rearrangement in the gelsolin(1–3)-actin complex correlates to the weakening or destabilisation of the complex, identifying the molecular signatures of programmed bio-disassembly as exemplified here by the LPA-triggered sequestration of actin from gelsolin(1–3). Such separation of the complex provides a means for the facile detection of ovarian cancer [17].

Identified as a critical gelsolin residue for LPA insertion in the current work, the potential role of Lys150 in binding pocket dynamics and molecular recognition could be further explored in future site-directed mutagenesis studies supported by long MD simulations with an isosteric in silico mutation of Lys150 in the PIP_2_-binding domain of gelsolin. Furthermore, as there are various species of LPA, this study can be expanded to model the binding of, for example, various acyl species of LPA molecules to gelsolin(1–3)-actin and compare their potential to disrupt the protein complex. Future studies of bio-sensing combining predicted and experimental binding assays with surface chemistry will further explore the applicability to detect other disease biomarkers. The proposed mechanism obtained in the current modelling study predicts a simple means of bio-complex destabilisation based on bridging amphipathic molecules across the protein–protein interface via simultaneous binding to hydrophilic and hydrophobic sites. This multi-site binding weakens the interface, triggering the disassembly and release of the binding partners for biosensing at experimental timescales.

## Figures and Tables

**Figure 1 biomolecules-13-01426-f001:**
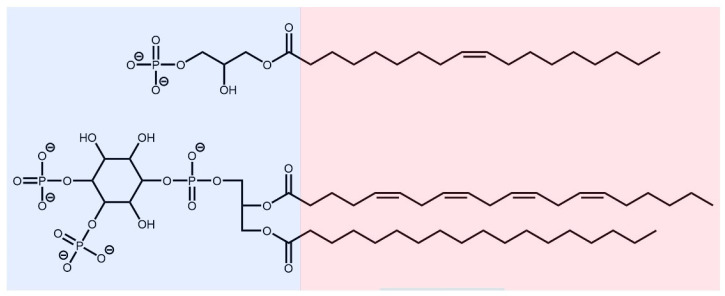
The polar and nonpolar regions of lysophosphatidic acid (LPA **above**) and phosphatidylinositol 4,5-bisphosphate (PIP_2_ **below**).

**Figure 2 biomolecules-13-01426-f002:**
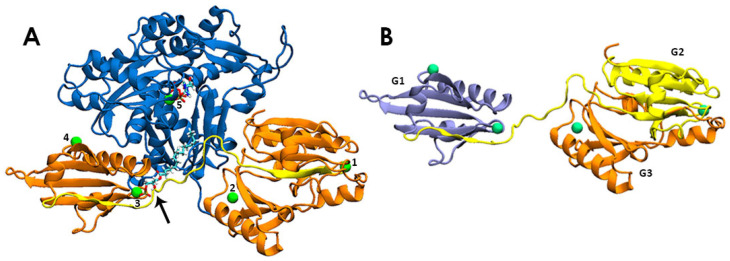
(**A**) The LPA-bound gelsolin(1–3)-actin complex: five numbered Ca^2+^ ions, actin (blue) and gelsolin(1–3) (orange, with the PIP_2_-binding domain coloured yellow). The arrow points to the LPA molecule, shown in stick representation, coordinated to Ca^2+^ #3. An ATP molecule binds Ca^2+^ #5. (**B**) The distinct domains of gelsolin(1–3), from left to right, G1 (violet), G2 (yellow), and G3 (orange).

**Figure 3 biomolecules-13-01426-f003:**
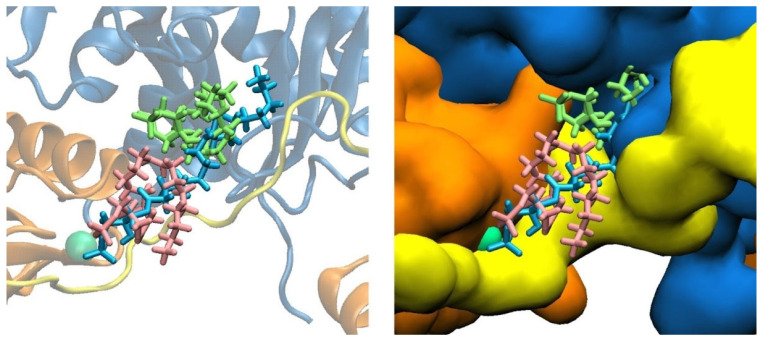
LPA docked to gelsolin(1–3)-actin in Dock 1 (pink), Dock 2 (green), and Dock 3 (light blue) models. Actin is dark blue, gelsolin is orange, and the PIP_2_-binding domain in gelsolin is coloured yellow in both the cartoon (**left**) and molecular surface representation (**right**) of the protein complex.

**Figure 4 biomolecules-13-01426-f004:**
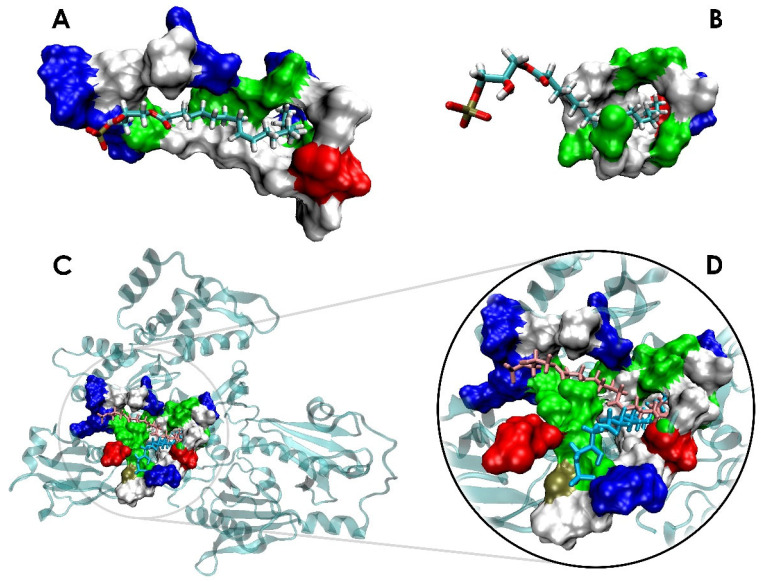
Actin residues within 3 Å of LPA following 2 µs of dynamics, defining the actin LPA-insertion pocket from two different starting structures: (**A**) Dock 2 and (**B**) Dock 3. (**C**) LPA binding to the gelsolin(1–3)-actin complex after 2 µs from the starting structures of Dock 2 (pink) and Dock 3 (light blue), with (**D**) the inset projecting on the designated LPA-insertion pocket. Residues that are within 3 Å of LPA are coloured based on polarity: nonpolar (white), polar (green), basic (blue), and acidic (red).

**Figure 5 biomolecules-13-01426-f005:**
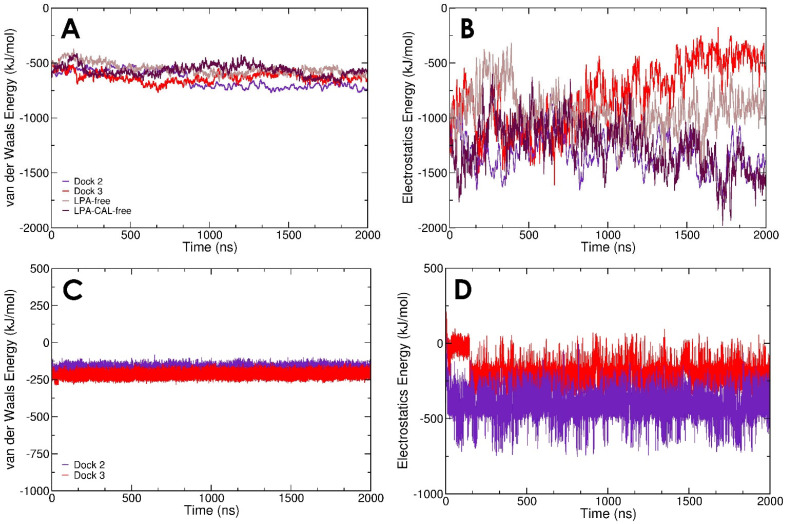
Timelines of van der Waals and electrostatic contributions to the interaction energies between (**A**,**B**) gelsolin(1–3) and actin and (**C**,**D**) the gelsolin(1–3)-actin complex and LPA.

**Figure 6 biomolecules-13-01426-f006:**
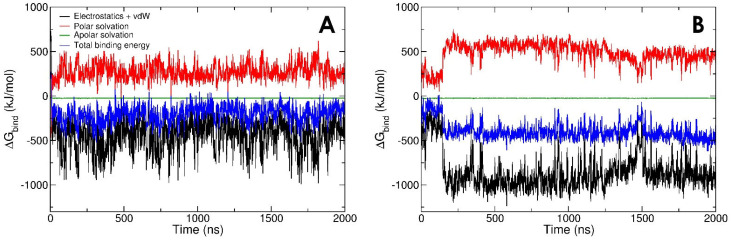
Timelines of binding energies of LPA with gelsolin(1–3)-actin in (**A**) Dock 2 and (**B**) Dock 3 models.

**Figure 7 biomolecules-13-01426-f007:**
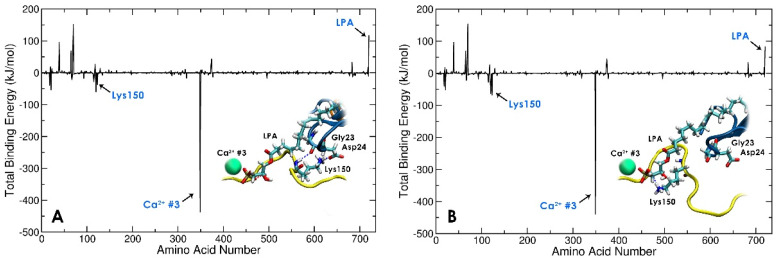
Residue-wise decomposition of binding energies in the gelsolin(1–3)-actin-LPA complex identifying the local conformational change of Lys150 in gelsolin(1–3), where (**A**) Lys150 is H-bonded with actin during early stage sub-200 ns dynamics and (**B**) Lys150 H-bonds with LPA in the last 200 ns of dynamics.

**Table 1 biomolecules-13-01426-t001:** Gelsolin(1–3) and Actin residues binding LPA.

Model	Residues within 3 Å at 0 µs	Residues within 3 Å at 2 µs
Dock 1	*Gelsolin(1–3)*: Glu92, Ser94, Gln95, Ser147 *, Phe149 *, His151 *	N/A
	*Actin*: Asp25	N/A
Dock 2	*Gelsolin(1–3)*: Gln95, Phe149 *	*Gelsolin(1–3)*: Gln95, Arg120
	*Actin*: Gly23, Asp24, Asp25, Ala26, Glu334, Tyr337, Trp340, Ile341	*Actin*: Lys18, Ala19, Gly20, Phe21, Ala22, Asp24, Asp25, Pro27, Ser145, Arg147, Ile330, Pro332, Glu334, Arg335, Tyr337, Ser338, Trp340, Ile341
Dock 3	*Gelsolin(1–3)*: Glu92, Ser94, Val145 *, Ala146 *, Ser147 *, Phe149 *, Lys150 *, His151 *	*Gelsolin(1–3)*: Glu92, Ser94, Gln95, Val145 *, Ala146 *, Ser147 *, Phe149 *, Lys150 *
	*Actin*: Val9, Asp11, Lys18, Asp24, Asp25, Ala26, Pro27, Trp340, Ile341	*Actin*: Val9, Asp11, Lys18, Gly20, Ala22, Gly23, Pro27, Tyr337, Trp340, Ile341, Ser344

* Residues belonging to the gelsolin(1–3) PIP_2_-binding domain.

## Data Availability

The data presented in this study are available in this article (and Appendix A).

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
