# Peer review of "Coupled Electrostatic and Hydrophobic Destabilisation of the Gelsolin-Actin Complex Enables Facile Detection of Ovarian Cancer Biomarker Lysophosphatidic Acid"

_biomolecules, 2023, doi:10.3390/biom13091426_

Round 1

Reviewer 1 Report

Davoudian and colleagues propose a model how LPA splits the gelsolin-actin complex using a highly sophisticated modeling- and molecular dynamics-based approach.  The objective of the study is to ultimately develop a biosensor for the lipid mediator lysophosphatidic acid (LPA).

The methodology is adequate and elegantly pursued in the triple complex.  The conclusion that model D3 dominated by a combination of charge- and van der Waals forces is supported with the findings in comparison with models D1 and D2.  The real strength of the approach is the MD simulation that reveals how the hydrophobic tail of LPA intercalates itself between the two proteins, in turn weakening the van der Waals forces leading to a dissociation of actin from gelsolin.

I have two major points of criticism:

1.     The authors arrive at the conclusion that Lys150in gelsolin is the critical residue in the insertion of LPA and leading to the dissociation of the two proteins.  Assuming that the authors are not wet-lab scientists, nevertheless, an isosteric in silico mutation of Lys 150 should be included to support its critical role in splitting the complex.

2.     I think, that the authors should have modeled the various acyl species of LPA if they are serious of developing the actin-gelsolin biosensor for diagnostic purposes.  Serum and plasma contain multiple LPA molecular species and knowing and ranking their “splitting” efficacy and binding free energies would aid the development of a diagnostic assay.  I think the above two experiment should be easily doable and adda lot of strength to the manuscript.

Minor comment:  Dr. YAn Xu's JAMA paper claiming that LPA was an early marker in  ovarian cancer has been disputed and discredited by several reports, including JAMA itself. Therefore, starting the story line with this paper is misleading and incorrect.  The main problem was in that study that that LPA in the peritoneal compartment  is not reflected by that of the plasma compartment.  Thus, measuring LPA levels is palsam can hardly be viewed as an "early marker of OC".  LPA now is viewed as a paracrine mediator - see the work of the Aoki group and the Perrakis group.

Author Response

Please view attachment

Reviewer 2 Report

This manuscript by Michael and Daniel Thompson is an interesting piece of work providing insight into binding of lysophosphatidic acid (LPA) in the gelsolin-actin complex. LPA binding in the gelsolin-actin complex weakens gelsolin-actin interactions leading to release of actin which can be used as the basis of possible biomarker for quantification of ovarian cancer. The authors have used three different software (AutoDock4, HDOC, and AutoDock Vina) to perform blind as well as biased molecular docking involving crucial amino acid residues to obtain energetically favorable docked poses of the LPA in the potential binding site. Authors have then gone ahead to perform extended molecular dynamic simulations for 10 microseconds to identify energetically favorable conformation over the course of simulation. They have convincing evidence to show that LPA unfolds to attain energetically favorable elongated conformation. The hydrophilic head group binds to the PIP2 binding domain of gelsolin while hydrophobic tail binds to a tunnel cavity of action through nonbonding interactions, including hydrogen bonding, electrostatic, and Van der Waals interactions. These interactions bring in conformational changes in the gelsolin-actin complex and weaken the complex resulting in release of actin. These findings are crucial for potential future work on biosensing and fit well for the readership in the Biomolecules. I would like to recommend inclusion of this manuscript upon minor corrections as listed below.

1.    It would be great to provide binding free energy (total H) from MD simulation for best docked poses (Dock 2 and Dock 3)

2.    It is very important to provide a table for residue wise energy decomposition to emphasize the crucial amino acid residues for binding. This is in addition to current Figure 7.

3.    There are two periods on line 404. One should be deleted.

Author Response

Please view attachment

Round 2

Reviewer 1 Report

The authors addressed my critique and suggestions by changing the text rather than doin additional experiments. The presentation of the filed is more balanced in the revised manuscript and the authors acknowledge that additional experimental work will be necessary to advance the model to a diagnostic test.